# Automatic Fine-Tuned Offline-to-Online Reinforcement Learning via Increased Simple Moving Average Q-value

## Abstract

Offline-to-online reinforcement learning starts with pre-trained offline models and continuously learns via interacting with the environment in online mode. The challenge of it is to adapt to distribution drift while maintaining the quality of the learned policy simultaneously. We propose a novel policy regularization method that aims to automatically fine-tune the model by selectively increasing the average estimated Q-value in the sampled batches. As a result, our models maintain the performance of the pre-trained model and improve it, unlike methods that require learning from scratch. Furthermore, we added efficient $\mathcal{O}(1)$ complexity replay buffer techniques to adapt to distribution drift efficiently. Our experimental results indicate that the proposed method outperforms state-of-the-art methods on the D4RL benchmark.

## 1 Introduction

Traditionally, training and evaluation for Reinforcement learning (RL) is conducted in an online fashion while interacting with the environment (Silver et al., 2017; Todorov et al., 2012; Mnih et al., 2013; Silver et al., 2014; Fujimoto et al., 2018; Haarnoja et al., 2018). However, in many real-world problems, it is inefficient or infeasible to build simulators or models of the environments. Learning from randomly initialized policy is risky and dangerous in many domains, e.g., healthcare, industrial control, and trading.

Batch or offline reinforcement learning methods (Levine et al., 2020; Lange et al., 2012) learn from p logged interactions which are stored as replay buffers (Lin, 1992). It requires no interaction with the environment during training, and it resembles most real-world use cases where there is existing data that could be treated as prior knowledge. In such settings, it is typical to have no accurate or reliable simulators of the environment. Exploration is limited for the offline approach because of the extrapolation errors induced by out-of-distribution (OOD) action selection. Thus, offline approaches tend to regularize the policy with behavioral policy or pessimistically underestimate the values (Fujimoto et al., 2019; Kumar et al., 2020; Fujimoto & Gu, 2021).

Offline-to-online (O2O) reinforcement learning could further improve the performance of the pre-trained offline model with online learning. Nonetheless, avoiding policy collapse and adapting to the distribution drift at the beginning of the transition is challenging. Many previous methods (Zheng et al., 2023; Lee et al., 2022; Zhang et al., 2023) could achieve better policies than their pre-trained offline models but suffer from policy collapse. It means the models could not maintain the performance of existing pre-trained offline models. Instead, they suffer a sudden decrease in performance or even learn from scratch at the beginning of the transition from offline to online. Safe RL methods (Laroche et al., 2019; Scholl et al., 2022) might be assumed to solve the aforementioned problem due to their safe policy improvement techniques. However, the key limitation is that all these papers assume the ability to interact with the environment and behavior policy to accurately estimate the baseline performance. In offline RL, we do not have access to interact with the environment or behavior policy. We only have a fixed batch of logged data. Without environmental interaction, we cannot reliably estimate the baseline performance. Therefore, these papers do not directly apply in an offline setting. New methods are needed to constrain policy updates without relying on accurate baseline estimates from the environment.

We propose a novel regularization method in policy learning to selectively maximize the difference between the mean value of the sampled batch to a previous mean reference value. The key insight of our method is to train the agent to yield a policy that generates transitions with a higher average Q-value as training continues when the value function network is not converging and learns like a conservative offline model when the value network is converging. We also adopt previously proposed methodologies in experience replay buffers: combined experience replay (CER) (Zhang & Sutton, 2017) to include the latest transition and remove the oldest policy (Fedus et al., 2020) by using a smaller replay buffer. We use a bootstrapped ensemble Q-network with an outlier filtering technique for more accurate value estimation to reduce the uncertainty encountered during the distribution drift.

To summarize, our contributions are:

- We design a novel regularization method in policy learning to automatically decide if we want to maximize the average Q-value of the replay buffer to accelerate the O2O learning with a value convergence constraint.

- We incorporate replay-buffer techniques in O2O to adapt to the distribution drift: CER and removing the oldest policy with $\mathcal{O}(1)$ costs.

- Q-network outlier filtering to stabilize the Q-value estimation in ensemble learning.

- Our method requires fewer assumptions and is more efficient. It requires no information on the expert or random agent performance, does not re-train offline models and does not require extra models except for the pre-trained offline model.

## 2 RELATED WORK

### 2.1 OFFLINE RL

In many real-world settings, we have access to data generated with an existing 'behavioral' policy when there are no established simulators of the environment. These logged interactions are saved as experience replay buffers. Offline RL learns exclusively from existing static datasets without interacting with an environment. Due to the lack of accurate value estimation of OOD actions, these methods learn a more conservative policy or a pessimistic lower bound of the true value function. BCQ (Fujimoto et al., 2019) mitigates the extrapolation errors induced by OOD actions via a variational autoencoder. BEAR (Kumar et al., 2019) uses ensemble Q-functions to reduce the accumulation of bootstrapping errors. BRAC (Wu et al., 2019) regularizes the learned policy towards the behavioral policy with a KL divergence constraint between the distributions over actions. CQL (Kumar et al., 2020) learns a lower bound of the true Q-function with SAC (Haarnoja et al., 2018)-style entropy regularization. TD3+BC (Fujimoto & Gu, 2021), derived from TD3 (Fujimoto et al., 2018), uses a behavioral cloning regularization for policy learning. UWAC (Wu et al., 2021) down-weights the OOD state-action pairs' contribution to the training. Swazinna et al. (2022) presents a method to let the user adapt the policy behavior after training is finished. Ghosh et al. (2022) proposes an adaptive offline method in a Bayesian sense involves solving an implicit POMDP (Partially Observed Markov Decision Process). In this study, we specifically concentrate on the evaluation and comparison of model-free reinforcement learning methods. Our aim is to delve into the performance, robustness, and scalability of model-free approaches in addressing the distribution drift while no models or simulators are built. While acknowledging the significance of model-based methods, we deliberately limit our investigation to model-free algorithms to provide a comprehensive understanding of their capabilities in isolation.

### 2.2 OFFLINE-TO-ONLINE RL

Offline-to-Online RL follows the assumption of offline RL where there is no access to the simulator of the system. However, we could further improve the model with online interactions since the pure offline method cannot yield accurate value estimation of the OOD state-action values. Hence, the goal is to enhance the capability of the model with online training without learning from scratch as in the traditional online setting.

### 2.2.1 RL WITH OFFLINE DATA

Previous studies focus on RL boosted with offline data. One branch in this research area is RL with Expert Demonstrations (RLED) with the assumption that a pre-trained offline model may not be necessary. APID (Kim et al., 2013) leverages few and/or sub-optimal demonstration data that is used as suggestions to guide the optimization performed by approximate policy iteration. DQfD (Hester et al., 2018) leverages demonstration data to accelerate the online learning process. Piot et al. (2014) proposes a method to minimize the optimal Bellman residual guided by constraints defined by the expert demonstrations. Recently, RLPD (Ball et al., 2023) extends standard off-policy RL and achieves state-of-the-art online performance on a number of tasks using offline data not limited to expert prior knowledge (Ball et al., 2023). This branch of research is different from our study in that we focus on fine-tuning the pre-trained offline models and not training the models from scratch with offline data to accelerate the learning.

### 2.2.2 ONLINE FINE-TUNING WITH OFFLINE PRE-TRAINING

Another branch, which is similar to our proposed method, assumes we fine-tune offline models in online settings to adapt to distribution drift. AWAC (Nair et al., 2020) trains an advantage-weighted actor-critic with an implicit policy constraint to avoid over-conservative behavior. Balanced replay is a method with a balanced replay between offline and online buffers, a pessimistic Q-ensemble (Lee et al., 2022), and a density ratio estimator to improve sample efficiency and prevent over-optimism. PEX (Zhang et al., 2023) freezes the pre-trained offline model, expands the policy set with the fine-tuning model, and constructs a categorial distribution for selecting the final action. APL (Zheng et al., 2023) obtains near-on-policy data and chooses an optimistic update strategy. On the other hand, it uses a pessimistic update strategy for sampled offline data. REDQ+ABC (Zhao et al., 2022) uses randomized ensemble Q-functions to increase sample efficiency and adaptive hyperparameter tuning to adjust the degree of behavioral policy regularization with a normalized target episode reward. ACA (Yu & Zhang) introduces a reconstruction of Q-functions for online fine-tuning as an alignment step so it is tamed to be consistent. TD3-C (Luo et al., 2023) considers conservative policy optimization as the approach for stabilizing finetuning when the offline dataset lacks diversity. (Hong et al., 2022) propose to learn a condition-adaptive policy that could adjust the degree of conservatism using online interaction.

Unfortunately, the aforementioned offline-to-online methods need at least one of the following requirements that makes them resource-consuming (Yu & Zhang; Zhao et al., 2022; Lee et al., 2022; Luo et al., 2023): Changing the offline training processes (requires re-training of the offline models), introducing additional models other than existing ones, and maintaining multiple buffers. Other methods require information on absolute scores of expert and random agents that may not be accessible. Many suffer policy collapse at the very beginning of the transition from offline mode to online mode. Our method requires fewer assumptions, is efficient (single replay buffer, no extra models), but still outperforms other methods.

## 3 BACKGROUND

RL problems are formulated as a Markov Decision Process (MDP), a sequential decision-making problem that aims to maximize the discounted accumulative rewards. The MDP consists of a tuple: $\mathcal{M} = (\mathcal{S}, \mathcal{A}, \mathbb{P}, r, \gamma)$, where $\mathcal{S}$ is the state space, $\mathcal{A}$ is the action space, $\mathbb{P}$ is the transition dynamics. The next state $s_{t+1} \sim p(\cdot|s_t, a_t)$ is decided by the current state and the action selected by a policy $\pi(a|s), \pi : \mathcal{S} \to \mathcal{A}$ either in a stochastic or a deterministic fashion. The reward function $R : \mathcal{S} \times \mathcal{A} \to \mathbb{R}, r \in \mathbb{R}$ is mapped as a scalar, and the discount factor $\gamma \in [0, 1)$. The agent's goal is to optimize the policy to maximize the discounted accumulated return $\mathbb{E}_\pi[\sum_{t=0}^{\infty} \gamma^t r_t]$ (Sutton & Barto, 2018).

### 3.1 OFFLINE TRAINING

In offline training, the replay buffer $\mathcal{D}$ is generated by an unknown behavioral policy (or a combination of multiple policies) $\pi_\beta(s)$. Then, the offline model aims to learn the optimal policy without interacting with the environment within the confined state-action visitations. Thus, when the trained

offline RL policy is deployed in the real environment, any OOD actions may lead to inaccurate value estimation due to extrapolation errors.

Our method builds on TD3+BC, which is an offline version of TD3 (Fujimoto et al., 2018) with minimal modification from its online version. The learned policy is regularized with a behavior cloning term:

$$\pi = \arg\max_{\pi} \mathbb{E}_{(s,a)\sim\mathcal{D}}[\lambda\bar{Q}(s,\pi(s)) - (\pi(s) - \pi_{\beta}(s))^2], \ \lambda = \frac{\alpha}{\frac{1}{N}\sum_{(s_i,a_i)}|Q(s_i,a_i)|} \tag{1}$$

Where N is the size of the minibatch, $\bar{Q}$ is the average Q-value in the sampled batch given $s$ and $\pi(s)$, $\pi_{\beta}(s)$ denotes the behavioral policy given $s$, and $\alpha$ is a hyperparameter to balance between the online exploration and the exploitation of the behavioral policy.

## 4 METHODOLOGY

### 4.1 INCREASED SIMPLE MOVING AVERAGE Q-VALUE

Our method ISMAQ (Increased Simple Moving Average of Q-value) is simple and straightforward – it aims to learn a policy that yields an increasing simple moving average (SMA) Q-value in the sampled batch compared to a previous reference SMA. We use SMA instead of the vanilla average since the average Q-value in the batch is noisy due to random sampling and inherited uncertainty within the models. The timestep difference between the current SMA and the reference SMA is a hyperparameter to be optimized; it depends on how rapidly the value estimation varies. If we simply use greedy Q-value increment, it would lead to abrupt performance fluctuations when Q-value estimation is uncertain while encountering unseen environment state-action distributions. Thus, we add safe constraints to conservatively train the models when observing decreased Q-SMA.

#### 4.1.1 OBSERVATION AND INSIGHT

To develop ISMAQ, we conduct preliminary experiments to gain some insights into the episodic average Q-value progression with fine-tuning of pre-trained TD3+BC models with online interactions, i.e. pre-trained TD3+BC models convert to TD3 online training.

Specifically, we add up all the averaged Q-values of the sampled batches at each timestep $i$ for $\bar{Q}_{\mathcal{B}_i}$. Then, when an episode ends at time $t_e$ we store the episodic average $\bar{Q}_e = \frac{1}{t_e}\sum_{i=1}^{t_e}\bar{Q}_{\mathcal{B}_i}$. Finally, we plot the average episodic mean Q-values over time in Fig. 1 (where the solid blue curves are the average values, and the shaded regions are the min/max range).

We observe that the average Q-value increases as training proceeds for buffers with lower behavioral policy performance (i.e. medium buffers). For the expert task, the model is unable to learn a better policy to yield a higher estimated Q-value during the training, i.e., the value estimation converges (Silver, 2015). This result is consistent with the mean episode return in the main experiments we shall present later.

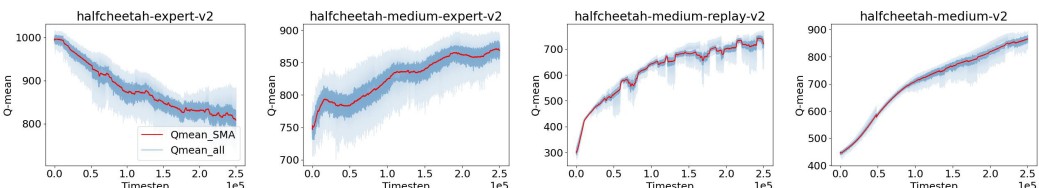

Figure 1: The average Q-value in the sampled batches in each episode of training.

#### 4.1.2 AVERAGED Q-VALUE IN REPLAY BUFFERS

Based on the observation in Sec. 4.1.1, we argue that a sub-optimal agent will yield a higher average Q-value estimate in the replay buffer with further online learning that leads to policy improvement. It could be proved by the following:

With policy improvement, the value function correspondingly improves based on the standard policy improvement theorem (see Appendix B).

**Lemma**: An improved policy will yield a higher randomly sampled average Q-value from the replay buffer.

***Proof***: Consider the Q-learning Watkins & Dayan (1992) update rule:

$$Q(s,a) \leftarrow Q(s,a) + \alpha(r + \gamma \max_{a'} Q(s',a') - Q(s,a))$$

where $Q(s,a)$ is the Q-value for the state-action pair (s,a), $\alpha$ is the learning rate, $r$ is the immediate reward, $\gamma$ is the discount factor, $s'$ the next state, $a'$ the next action. $\pi'$ is an improved policy over $\pi$. The improved policy $\pi'$ selects actions that, on average, lead to higher expected returns.

$$Q^{\pi'}(s,a) = Q^{\pi}(s,a) + \alpha(r + \gamma \max_{a'} Q^{\pi'}(s',a') - Q^{\pi}(s,a))$$

$$\max_{a'} Q^{\pi'}(s',a') \geq \max_{a'} Q^{\pi}(s',a')$$

$$Q^{\pi'}(s,a) \geq Q^{\pi}(s,a)$$

It implies that an improved policy $\pi'$ will yield a higher expected Q-value than the original policy ($\pi$) for the state-action pair $(s,a)$. The proof holds for any randomly sampled state-action pair from the replay buffer, demonstrating that the improved policy results in higher randomly sampled average Q-values.

### 4.1.3 ISMAQ IMPLEMENTATION

**Auto-tuned ISMAQ**

As we can observe in Fig. 1, the trace of the episodic average Q-value is noisy while the Q-SMA is more stable. Thus, we introduce the simple moving average of the average Q-values to yield a more statistically meaningful metric for the model. Our method uses the pre-trained TD3+BC models as the initialized policy without the requirement to modify the offline training. To apply our method on TD3+BC, we modify the policy update from equation 1 with the added loss term $\mathcal{L}_{ISMAQ}$:

$$\mathcal{L}_{ISMAQ} = ReLU\left(\frac{\bar{Q}_{SMA}^t - \bar{Q}_{SMA}^{t-d}}{\bar{Q}_{SMA}^t + \bar{Q}_{SMA}^{t-d}}\right) \tag{2}$$

where $t$ is the current timestep, and $d$ is the difference between the current timestep and the reference timestep. ReLU is the rectified linear unit activation function which automatically tunes this term based on the difference of the Q-SMA between the reference timestep and current timestep and our ensemble-Q:

$$\bar{Q}(s,\pi(s)) = \frac{1}{2K}\sum_{j=1}^{2}\sum_{i=1}^{K} Q_{i,j}(s,\pi(s)) \tag{3}$$

where $i$ is the $i$th ensemble-Q in $K$, and $j$ is the $j$th Q-network in double Q-network of TD3. And the SMA for the timestep $t$:

$$\bar{Q}_{SMA_t} = \frac{\bar{Q}_t + \bar{Q}_{t+1} + ... + \bar{Q}_{t+w}}{w} \tag{4}$$

where $w$ is the window size we use for calculating the SMA. Thus, the policy update follows:

$$\pi = \arg\max_{\pi} \mathbb{E}_{(s,a)\sim\mathcal{D}}[\lambda\bar{Q}(s,\pi(s)) - (\pi(s) - \pi_{\beta}(s))^2 + \xi\mathcal{L}_{ISMAQ}] \tag{5}$$

where $w$ is the window size for calculating the SMA, and $\xi$ is a hyper-parameter for co-efficient of $\mathcal{L}_{ISMAQ}$. With the ReLU activation function, the added loss term is bounded, i.e. $\mathcal{L}_{ISMAQ} \in [0,1]$.

In Fig. 2, the black curve represents ISMAQ with only greedily adding the exploration term (Eq. 2 without ReLU). It degrades the model performance when unseen state-action distribution is encountered. However, by adding the ReLU as a safe constraint in the exploration term, we could conservatively maintain the current performance of the agent to yield a more reliable and stable policy.

**Q-network Outlier Filtering**
We also observe that the weak models in the Q-ensemble substantially harm the models' performance during the training (see Fig. 3). Thus, we impose a constraint to remove the outlier during the policy update: First, we get the average of the Q-value estimates among the ensemble Q-networks. Then, we find the models with the largest absolute difference between the mean and itself. Finally, we exclude the particular model in the policy update [1]. As a result, the policy training is more robust after the filtering (see Fig. 4, detailed description is in Appendix C).

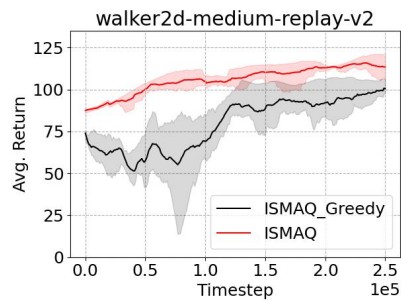

Figure 2: Adding the ReLU activation helps stabilize the model training

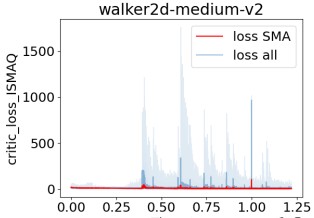

Figure 3: Outlier Q-network impacts on the critic losses

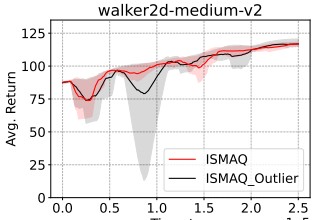

Figure 4: Comparison between Q-ensemble models with and without the outlier.

**To summarize**, ISMAQ optimizes the policy by identifying if the current SMA of the averaged Q-value is higher than a previous reference SMA. If so, we encourage the policy to maximize this loss term. Otherwise, when the current SMA is lower than the reference value, we keep the actor loss as is since it means the Q-value is converging, i.e., leave it learning as the original offline model, which is conservatively trained with online transitions.

### 4.2    Adapting to Distribution Drift

Another critical challenge in O2O transition is the distribution drift. Previously, several methods are proposed to accelerate RL learning by utilizing replay buffers (Zhang & Sutton, 2017; Schaul et al., 2016; Fedus et al., 2020). We found some of them are suitable to deal with the distribution drift in the O2O setting. The first one we adopt is the combined experience replay (CER) (Zhang & Sutton, 2017), which adds the latest transition to the sampled batch and could speed up the learning. Intuitively, it forces the model to learn from the latest state-action distribution of the environment combined with the previous ones. The other technique we adopt is removing the oldest transition in the buffer faster by setting a smaller number of transitions stored in the replay buffer since it is implemented in queues (Fedus et al., 2020). When a policy is learning and improving, the transitions generated by the old policies might harm the convergence of the model due to its inferior performance cf. the current policy. Especially in off-policy settings, we learn from the behavioral policy via replay buffer. Observations from our experiments (Sec. 5.3) indicate that the performance of the models consistently improves with the reduced age of the oldest policy (Fedus et al., 2020). These two techniques both could be implemented with minimal changes with only $\mathcal{O}(1)$ time complexity without modifying the algorithm itself, which is efficient and reasonable in O2O training. We detail all the steps of our method in Algorithm 1.

### 4.3    Ensemble Learning

Due to the need for exploring uncharted state-action spaces, most previous O2O studies take advantage of certain kinds of ensemble learning. However, most previous methods require random initialization for ensemble models to leverage. It is time-consuming to re-train the offline model and not reasonable to learn from scratch when pre-trained model is available. Thus, we use a more efficient method by bootstrapping $K$ ensemble double-Q networks via different combinations of the randomly sampled

---

[1]number of models excluded, $k_e$ could be tuned as a hyperparameter, in our experiments we use $k_e = 1$

batches in each iteration. To utilize the nature of the double Q-learning inherited in TD3+BC, we have a total of $2K$ estimated average Q-values in each training iteration and then average over them as the final value estimation for policy training (see Eq. 3).

# 5 EVALUATION

## 5.1 MAIN RESULTS

The main goal of offline-to-online RL is to maintain the pre-trained models' performance and continuously improve it as online training proceeds. We conduct the benchmark experiments with several state-of-the-art methods in MuJoCo (Todorov et al., 2012) control tasks with OpenAI gym (Brockman et al., 2016) environments. We compare our method with the following algorithms including the baseline method transitioning to online in different settings:

- **REDQ+ABC** (Zhao et al., 2022): It combines the randomized ensemble Q-functions to improve sample efficiency along with a proportional-derivative (PD) controller to tune the hyperparameter of the weight of the behavioral cloning term $\alpha$ in Eq. 1 with a target score and current episodic return.

- **Balanced Replay** (Lee et al., 2022): It trains an ensemble of pairs of CQL (Kumar et al., 2020) actor-critic agents with a prioritized buffer mixed with online and offline buffers and density ratios models.

- **TD3+BC to TD3** 1 to TD3 (Fujimoto et al., 2018): With the baseline method, we directly convert the offline TD3+BC to TD3 by removing the behavior cloning term shown in Eq. 1 at the beginning of the offline-to-online training.

- **PEX** (Zhang et al., 2023): A policy expansion approach that adaptively composes a frozen behavioral policy and a learnable policy.

As we observe in Fig. 5, our method ISMAQ not only maintains the proficiency of the pre-trained models but also improves upon them as training advances. From the experimental results in Table 1, our method outperforms other state-of-the-art methods in a sum of the first-10 and the last-10 evaluation scores over 12 tasks. [2] It also empirically demonstrates that in most of the tasks, the model is still learning when we interrupt our experiments at 250K steps as shown in Fig. 1. While *walker2d-expert* and *walker2d-medium-expert* could be possibly improved, however, for other methods, they failed to learn a better policy. In the case of *hopper-expert*, the model performance saturates, but ISMAQ successfully maintains policy performance while other methods fail to do so.

Table 1: Normalized scores averaged with 4 random seeds with the first-10 and the last-10 evaluation scores and overall sum, the left column of each algorithm indicates the first-10 scores, and the right column shows the last-10. Bold font highlights scores with the highest among all. (hc:halfcheetah, ho:hopper, wa:walker2d, e:expert, m:medium, r:replay, all on D4RL v2.)

| Task | ISMAQ | | REDQ+ABC | | PEX | | Balanced Replay | | TD3+BC_TD3 | |
|---|---|---|---|---|---|---|---|---|---|---|
| hc-e | **95.2** | 99.4 | 93.6 | **101.7** | 34.9 | 93.5 | 11.7 | 75.9 | 13.8 | 92.6 |
| hc-m-e | 79.5 | 99.4 | **91.7** | **104.4** | 49.3 | 90.3 | 63.6 | 100.4 | 54.8 | 96.6 |
| hc-m | 56.0 | 93.3 | 50.1 | 99.4 | 47.7 | 65.4 | **71.3** | **99.9** | 58.6 | 89.9 |
| hc-m-r | 51.2 | 83.0 | 46.3 | 95.6 | 45.5 | 54.3 | **63.4** | **96.4** | 50.3 | 78.9 |
| ho-e | **111.0** | **111.6** | 101.3 | 109.3 | 18.8 | 66.1 | 31.2 | 93.3 | 43.7 | 107.9 |
| ho-m-e | 86.7 | **111.9** | **90.6** | 101.3 | 32.0 | 75.2 | 35.1 | 98.7 | 72.1 | 108.6 |
| ho-m | 90.7 | 103.7 | 59.8 | **110.8** | 35.5 | 92.0 | **93.1** | 103.0 | 86.8 | 105.5 |
| ho-m-r | **90.7** | **110.0** | 68.4 | 106.7 | 61.7 | 89.8 | 12.2 | 29.4 | 85.5 | 109.5 |
| wa-e | 103.7 | **124.4** | **110.2** | 115.1 | 23.5 | 101.5 | 3.9 | 73.1 | 15.5 | 109.0 |
| wa-m-e | **111.9** | **123.0** | 110.5 | 112.1 | 71.5 | 110.0 | 4.57 | 89.4 | 48.7 | 115.4 |
| wa-m | **88.1** | 116.6 | 82.5 | **117.4** | 63.9 | 85.6 | 5.3 | 82.5 | 39.4 | 98.6 |
| wa-m-r | **88.2** | **113.8** | 80.3 | 112.7 | 74.4 | 88.6 | 73.4 | 101.2 | 69.7 | 100.3 |
| Sum | **1053.5** | **1289.9** | 986.0 | 1286.5 | 558.9 | 1012.3 | 468.7 | 1043.4 | 638.7 | 1212.7 |
| | **2343.4** | | 2272.5 | | 1571.2 | | 1512.1 | | 1851.4 | |

---

[2]In the "Sum" row of Table 1, the scores before "/" show the sum of first-10 and last-10, and the scores after "/" show total improvement from the pre-trained model's scores.

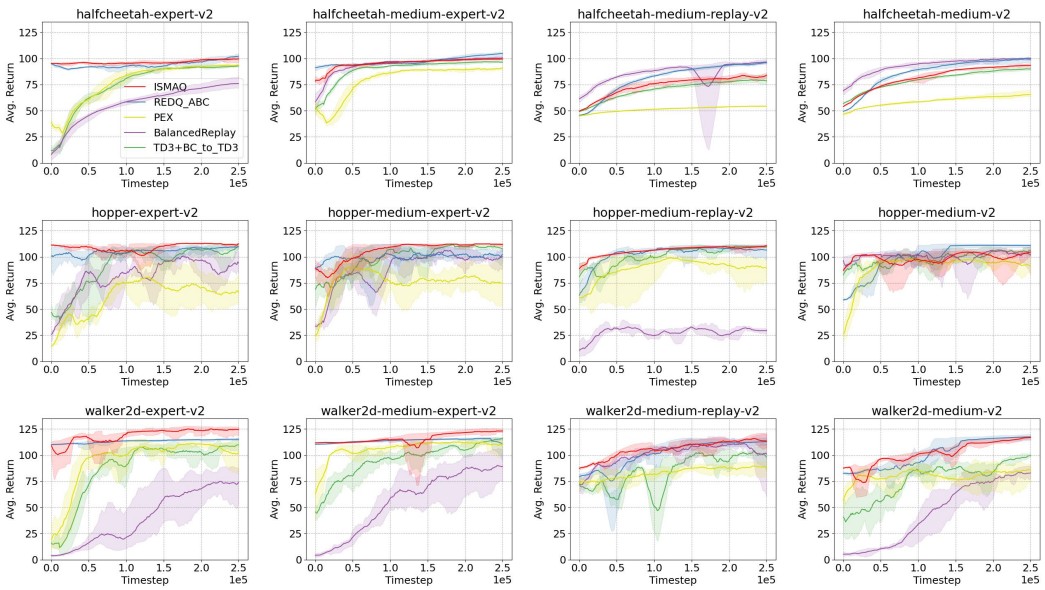

Figure 5: Learning curves comparison: All scores are normalized with expert policy as 100 and random policy as 0 standardized by D4RL (Fu et al., 2020) dataset. Solid lines and shaded regions represent mean and range, respectively.

## 5.2 SENSITIVITY ANALYSIS

We experiment on how ISMAQ's hyperparameter variation affects the model performance. In our method, there are three hyperparameters:[3]

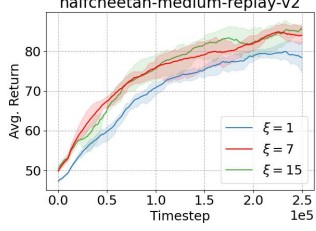

Figure 6: Weight of ISMAQ exp.

Figure 7: Number of ensemble-Q exp.

**The weight $\xi$ of the $\mathcal{L}_{ISMAQ}$ term** in policy learning (Eq. 5) controls how greedily the model should increase the averaged Q-values. Intuitively, we assume it is beneficial to set it to be larger for low-quality buffers and smaller for high-quality buffers. In Fig. 6 it shows that with a smaller weight of the ISMAQ term ($\xi$=1), the learning is slower compared with other values, while the largest one ($\xi$=15) might be beneficial to medium-replay buffer but harmful to expert buffer. We set $\xi = 7$ in all our experiments.

**The number of ensemble models $K$** in Eq. 3 decides how many Q-networks are trained in the value estimation. It affects the consumed computation resources and the variance between runs. Changing the number of the ensemble model does not necessarily affect the model's performance. However, with more models to decide the value estimation, the variance of the model's score is smaller. (see Fig. 7).

**The timestep difference $d$** from the current timestep $t$ to the previous reference point $t-d$ is described in Eq. 2. It might depend on the speed of how Q-value increments in the environment. But it is

---

[3]We keep the window size of SMA as the same as $d$ throughout all the experiments.

mainly affected by the second derivative of the mean Q-value indicated in Fig. 1. Since the distance to the reference point might vary, the reference value is also calculated with SMA. Thus, the current value and the reference value are both Q-SMA. Both of them are stable with an optimized window size for SMA. Hence, the adjustment of the hyperparameter d will not affect the model performance significantly. Thus, in Fig. 8 shows the difference between varied $d$ is not obvious.

### 5.3 ABLATION EXPERIMENT

We conduct a series of experiments to show how each specific add-on influences learning. Our design of the experiment is to remove one enhancement at a time to compare to our full ISMAQ implementation and demonstrate the difference between the ablated methods. We chose the medium buffer since it is generated by a more monotonic and inferior policy than others. As the experimental results shown in Fig. 9, we could observe that without ISMAQ, the learning is the slowest among all other methods, especially for the *hopper-medium* environment. Then deleting old policies, as concluded by (Fedus et al., 2020), reducing the age of the oldest behavioral policy generally improves the performance of the off-policy models. The difference between removing CER and the ensemble model is not obvious, however, empirically it indicates they are both beneficial to the models' performance overall.

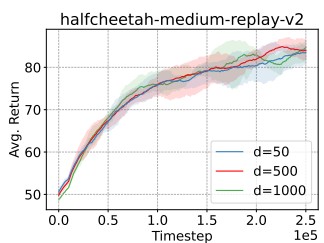

Figure 8: Timestep difference exp.

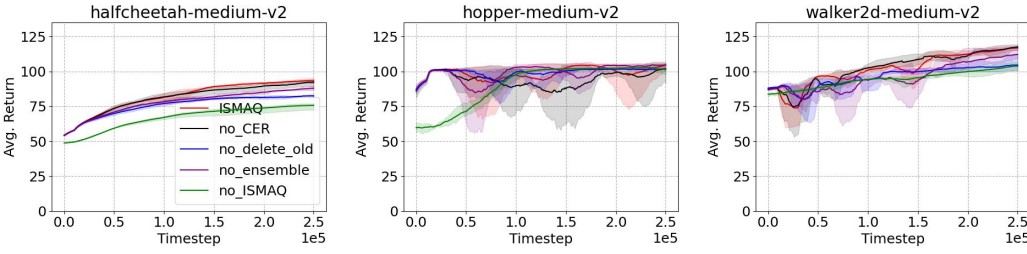

Figure 9: Ablation experiment: We remove the enhancement one at a time to observe how each one affects the model's performance. (no ISMAQ means the loss term of Eq. 2 is removed.)

## 6 CONCLUSION AND DISCUSSION

We propose a novel policy regularization method maximizing the simple moving average of the mean Q-value in the sampled batch in each timestep of training, with low-cost experience replay techniques to adapt distribution drift and improve sample efficiency with Q-ensemble. Extensive experimental results indicate that our method ISMAQ outperforms other state-of-the-art methods in the early offline-to-online transitions and the final learning scores.

The limitation of our method is that it relies on the accuracy of the Q-value prediction. Also, the initial policy's performance also limits our model's capability. However, we argue if the pre-trained models are not well- performed and/or the replay buffer is generated by ill-performed agents, it is unreasonable to fine-tune with these models. Instead, we should train online models from scratch. Additionally, ISMAQ has not been evaluated with stochastic MDPs. The MuJoCo benchmarks only use deterministic state transitions. The stochastic selection of initial states is not sufficient to be considered as stochastic MDPs (Mannor & Tamar, 2023). With our current work, we use a simple bootstrapped Q-ensemble to calculate the average based on all the ensemble's predictions. To improve from this, the distributional method could be applied and/or with critic losses' confidence level as uncertainty penalties. We will open-source our codes for research purposes.

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

# APPENDIX

## A    EXPERIMENT DETAILS

In this section, we record the software configuration, algorithm implementation, hardware configuration, training/evaluation details, and the hyperparameters settings in our experiments for reproducibility.

- **Software**
    - **Python**: 3.9.12
    - **Pytorch**: 1.12.1+cu113 (Paszke et al., 2019)
    - **Numpy**: 1.23.1 (Van Der Walt et al., 2011)
    - **CUDA**: 11.2

- **Algorithm implementation**
    - **TD3**: Author-provided implementation
    - **TD3+BC**: Author-provided implementation
    - **REDQ+ABC**: Author-provided implementation
    - **Balanced Replay**: Author-provided implementation
    - **CQL**: d3rlpy
    - **PEX**: Author-provided implementation

- **Hardware**
    - **CPU**: Intel Xeon Gold 6230 (2.10 GHz)
    - **GPU**: NVidia RTX A6000

- **Training and evaluation details**
    - **Offline pre-training**: 1M timesteps
    - **Online fine-tuning**: Training: 250K timesteps, evaluation frequency: 1K, number of evaluation episodes: 10
    - **Replay buffer sizes**: Downsample D4RL's original sizes with $5\%$ following the REDQ+ABC's (Zhao et al., 2022) settings.

## A.1 PERFORMANCE OF PRE-TRAINED OFFLINE MODELS

Table 2: Average scores of pre-trained offline models.

| Task/Algo. | ISMAQ | REDQ_ABC | PEX | Off2OnRL | TD3+BC_TD3 |
|---|---|---|---|---|---|
| halfcheetah-expert-v2 | 97.6 | 96.9 | 64.9 | -1.9 | 97.6 |
| halfcheetah-medium-expert-v2 | 93.7 | 95.4 | 74.1 | 37.2 | 93.7 |
| halfcheetah-medium-v2 | 48.0 | 48.7 | 41.8 | 58.2 | 48.0 |
| halfcheetah-medium-replay-v2 | 45.2 | 44.1 | 43.9 | 68.7 | 45.2 |
| hopper-expert-v2 | 111.5 | 109.6 | 14.0 | 16.9 | 111.5 |
| hopper-medium-expert-v2 | 80.1 | 99.9 | 5.5 | 94.9 | 80.1 |
| hopper-medium-v2 | 61.1 | 53.6 | 13.6 | 1.8 | 61.1 |
| hopper-medium-replay-v2 | 49.6 | 81.5 | 76.3 | 75.7 | 49.6 |
| walker2d-expert-v2 | 110.4 | 115.9 | 22.1 | 6.8 | 110.4 |
| walker2d-medium-expert-v2 | 110.8 | 116.3 | 67.9 | 0.1 | 110.8 |
| walker2d-medium-v2 | 82.1 | 78.8 | 74.9 | 87.3 | 82.1 |
| walker2d-medium-replay-v2 | 84.1 | 72.8 | 57.3 | -0.1 | 84.1 |
| Sum | 974.4 | 1013.5 | 556.1 | 445.4 | 974.4 |

## A.2 ISMAQ ALGORITHM

---

**Algorithm 1:** ISMAQ online fine-tuning

---

Load pre-trained offline model as $K$ ensemble double-Q networks $\{Q_{i,\theta_1}, Q_{i,\theta_2}\}_{i=1}^{K}$, actor network $\pi_\phi$,
  with random parameters $\{\theta_{i,1}, \theta_{i,2}\}_{i=1}^{K}, \phi$, target networks $\{\theta'_{i,1} \leftarrow \theta_{i,1}, \theta'_{i,2} \leftarrow \theta_{i,2}\}_{i=1}^{K}, \phi' \leftarrow \phi$, policy
  update frequency $f$, horizon $T$, replay buffer $\mathcal{B}$

**for** $t = 0$ *to T* **do**
  Select actions with exploration noise
  $a \sim \pi_\phi(s) + \epsilon, \epsilon \sim \mathcal{N}(0, \sigma)$
  Observe reward $r$ and next state $s'$
  Store transition $t = (s, a, r, s')$
  Delete the oldest one in $\mathcal{B}$ (Remove the oldest policy)
  **for** $i = 1$ *to K* **do**
    Sample $N$ transitions $(s, a, r, s')$ from $\mathcal{B}$ in which $t \subseteq \mathcal{B}$ (CER)
    $\tilde{a} \leftarrow \pi_{\phi'}(s') + \epsilon, \epsilon \sim \text{clip}(\mathcal{N}(0, \tilde{\sigma}), -c, c)$
    $y \leftarrow r + \gamma \min_{j=1,2} Q_{\theta'_{i,j}}(s', \tilde{a})$
    Update critics
    $\theta_{i,j} \leftarrow \arg\min_{\theta_{i,j}} N^{-1} \sum (y - Q_{\theta_{i,j}}(s, a))^2$
  **if** $t \bmod f$ **then**
    Update $\phi$ by policy gradient:
    Policy update follows Eq. 2, 3, 4, and 5(Ensemble-Q and ISMAQ)
    Calculate $\nabla_\phi J(\phi)$
    Update target networks:
    **for** $i = 1$ *to K* **do**
      $\theta'_{i,j} \leftarrow \tau\theta_{i,j} + (1 - \tau)\theta'_{i,j}$
    $\phi' \leftarrow \tau\phi + (1 - \tau)\phi'$

---

## A.3 GENERALIZATION EXPERIMENT

To further study how to generalize our approach in different settings, we conduct the experiments
to add ISMAQ in a pure online setting (see also Appx. D.1 and D.2 for pure offline and CQL
experiments). The experimental results shown in Fig. 10 indicate that combining ISMAQ with CER
and deleting old policy (TD3_ISMAQ_All[4]) could outperform TD3 in all three environments and
is more efficient than only adding ISMAQ. Without including the latest transition in the sampled
batch during training, the algorithm does not use information from the value estimated by the value
network and the actions selected by the policy network.

---

[4]For fair comparison, we do not include ensemble-Q in this experiment.

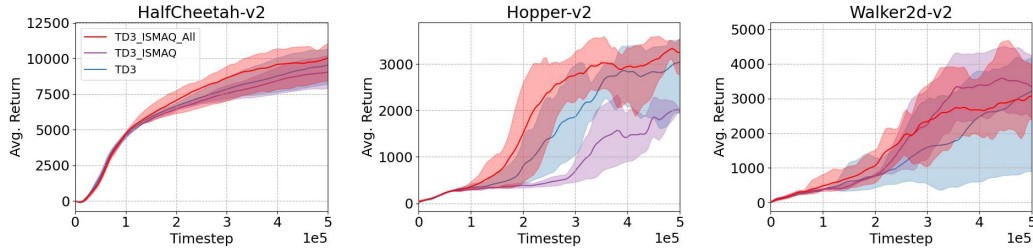

Figure 10: Online experiment: We add our enhancements onto TD3 in online training, and compare to its baseline method - TD3.

## B   POLICY IMPROVEMENT THEOREM

***Proof** (Silver, 2015)*:

Given a policy $\pi$

$$v_\pi(s) = \mathbb{E}[R_{t+1} + \gamma R_{t+2} + ...|S_t = s]$$

$\pi' = greedy(v_\pi)$. Consider a deterministic policy $a = \pi(s)$, improve it by acting greedily with respect to $v_\pi$

$$\pi'(s) = \arg\max_{a \in \mathcal{A}} q_\pi(s, a)$$

it improves the value from any state $s$ over one step:

$$q_\pi(s, \pi'(s)) = \max_{a \in \mathcal{A}} q_\pi(s, a) \geq q_\pi(s, \pi(s)) = v_\pi(s)$$

Hence, the value function is improved, i.e., $v_{\pi'}(s) \geq v_\pi(s)$

$$
\begin{aligned}
v_\pi(s) \leq q_\pi(s, \pi'(s)) &= \mathbb{E}_{\pi'}[R_{t+1} + \gamma v_\pi(S_{t+1})|S_t = s] \\
&\leq \mathbb{E}_{\pi'}[R_{t+1} + \gamma q_\pi(S_{t+1}, \pi'(S_{t+1}))|S_t = s] \\
&\leq \mathbb{E}_{\pi'}[R_{t+1} + \gamma R_{t+2} + \gamma^2 q_\pi(S_{t+2}, \pi'(S_{t+2}))|S_t = s] \\
&\leq \mathbb{E}_{\pi'}[R_{t+1} + \gamma R_{t+2} + ...|S_t = s] = v_{\pi'}(s)
\end{aligned}
$$

## C   Q-NETWORK OUTLIER FILTERING

During the policy training, for each $k$ Q-network in all $K$ ensemble models, we use the sampled $s$ and the selected actions $a = \pi(s)$ to get the estimated Q-values $Q_k(s, a)$. Thus, we could calculate the mean Q-value among all the ensemble models to get $\bar{Q}(s, a)$. Then we use $\arg\max$ to get the model with the maximum absolute difference from the mean and exclude that model, i.e. $\arg\max_k\{\sum_{i=1}^{N} |\bar{Q}_{k_i}(s, a) - Q_{k_i}(s, a)|\}$, where $N$ is the size of minibatch.

## D   ADDITIONAL GENERALIZATION EXPERIMENTS

### D.1   CQL + ISMAQ

To test if the generalization of ISMAQ we add our enhancement on another representative offline-RL method: CQL (Kumar et al., 2020). We conduct experiments on CQL-DB. It follows the CQL training but with dynamic buffers. And CQL-ISMAQadded on CQL-DB. We directly add our loss term in Eq. 2 on their actor loss as the following equation:

$$\phi_t := \phi_{t-1} + \eta_\pi \mathbb{E}_{s \sim \mathcal{D}, a \sim \pi_\phi(\cdot|s)} [Q_\theta(s, a) - \log \pi_\phi(a|s) + \xi \mathcal{L}_{ISMAQ}] \tag{6}$$

The experimental result in Fig. 11 indicates that ISMAQ improves CQL's in general. However, the amount of improvement is limited. Given the stochasticity nature of CQL, during the training the agent might generate a set of more explorative transitions than deterministic ones. Thus, the average Q-value in the buffer might not be a well-defined metric combined with ISMAQ.

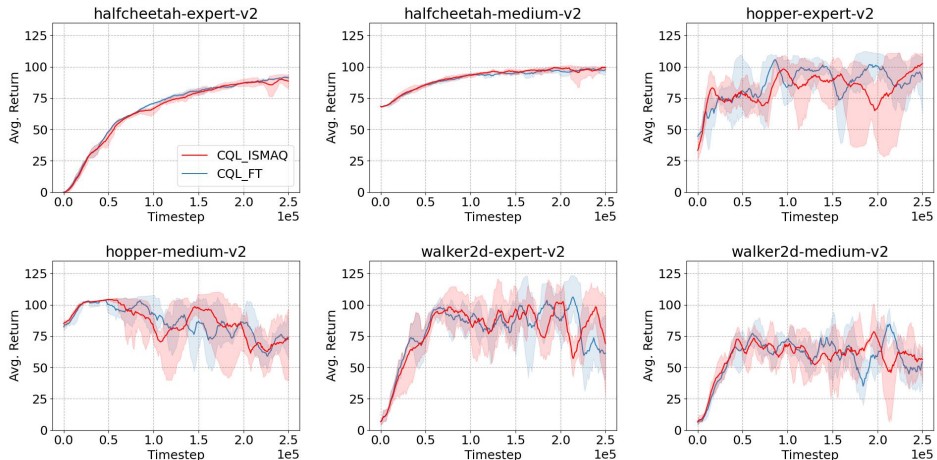

Figure 11: Experiment with CQL finetuning and CQL with ISMAQ

| Task | CQL_ISMAQ_DB | CQL_DB |
|------|--------------|--------|
| hc-e-v2 | 1.74 89.34 | **2.34 91.35** |
| ho-e-v2 | 47.45 **100.48** | **52.43** 93.86 |
| w-e-v2 | **11.61 81.85** | 11.43 63.08 |
| hc-m-v2 | **68.88 98.8** | 68.86 97.29 |
| ho-m-v2 | **87.57** 71.25 | 85.26 **73.82** |
| w-m-v2 | **10.05 56.75** | 8.73 49.77 |
| Sum | **725.77** | 698.22 |

### D.2 OFFLINE EXPERIMENT

We conduct the experiments to add ISMAQ into pure offline training and as expected there is only negligible improvement cf. the baseline method TD3+BC. Since the buffer is static, there is no exploration involved to improve the performance substantially (see Fig. 12)

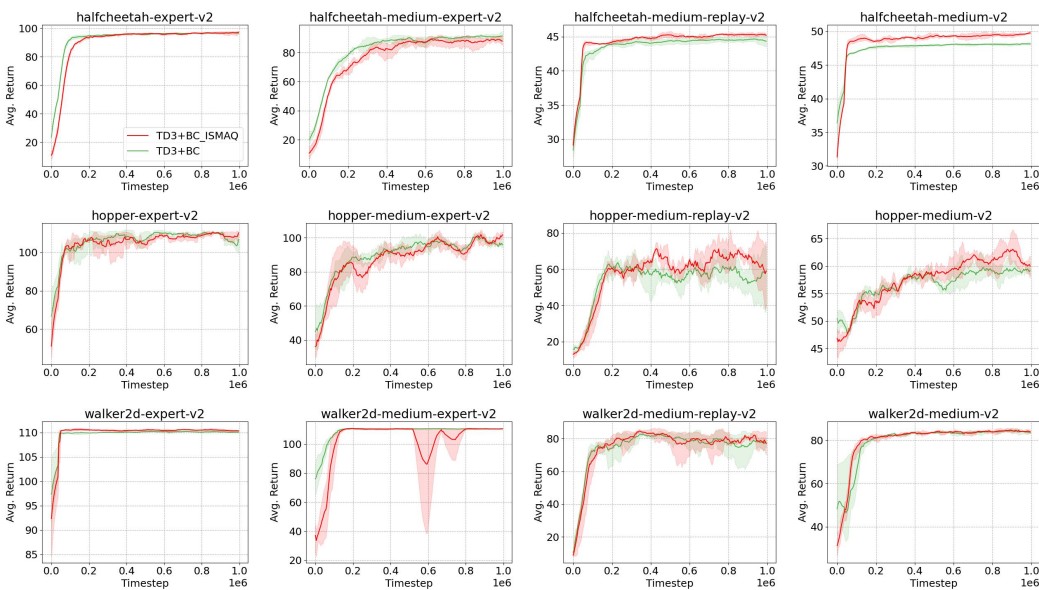

Figure 12: Offline experiment: We add our ISMAQ onto TD3+BC in offline training, and compare it with the baseline method TD3+BC, averaged over 3 seeds for both methods.

Table 3: Offline experiment score table

| Task | TD3_BC_ISMAQ | TD3_BC |
|------|--------------|--------|
| hc-e-v2 | **96.88** | 96.7 |
| hc-m-e-v2 | 88.72 | **91.08** |
| hc-m-r-v2 | **45.3** | 44.47 |
| hc-m-v2 | **49.66** | 48.15 |
| ho-e-v2 | **108.4** | 104.98 |
| ho-m-e-v2 | **99.7** | 96.13 |
| ho-m-r-v2 | **60.97** | 56.81 |
| ho-m-v2 | **60.3** | 59.2 |
| wa-e-v2 | **110.37** | 110.09 |
| wa-m-e-v2 | 110.41 | **110.45** |
| wa-m-r-v2 | **78.3** | 77.68 |
| wa-m-v2 | **84.2** | 83.99 |
| Sum | **993.21** | 979.73 |

## D.3 ANTMAZE AND ADROIT EXPERIMENTS

We have also conducted experiments on Antmaze and Adroit tasks as a reference. PEX demonstrates a higher score in these tasks. However, with all the MuJoCo tasks considered, ISMAQ still outperforms other benchmarks.

Table 4: Normalized scores of Antmaze and Adroit environments

| Task/Algo. | ISMAQ | PEX | REDQ_ABC | TD3+BC_TD3 |
|------------|-------|-----|----------|------------|
| pen-human-v1 | -3.4 | 90.0 | -2.1 | -3.0 |
| relocate-human-v1 | -0.3 | 2.9 | -0.3 | -0.3 |
| hammer-human-v1 | 0.2 | 2.1 | 0.3 | 0.3 |
| antmaze-large-diverse-v2 | 0.0 | 1.4 | 0.0 | 0.0 |

## E  MODEL PARAMETERS

We list the hyperparameters used (for TD3, TD3+BC, and our ISMAQ) in this paper for reproducibility. We keep the original hyperparameters setups as the authors' implementations since DRL methods are sensitive to hyperparameter tuning (Henderson et al., 2018) (see Table 5).

Table 5: ISMAQ, TD3, TD3+BC hyperparameters

| | Hyperparameter | Value |
|---|---|---|
| | Optimizer | Adam (Kingma & Ba, 2014) |
| | Critic learning rate | $3e^{-4}$ |
| | Actor learning rate | $3e^{-4}$ |
| | Mini-batch size | 256 |
| | Discount factor | 0.99 |
| Algorithm hyperparameters | Target update rate | $5e^{-3}$ |
| | Policy noise | 0.2 |
| | Policy noise clipping | (-0.5, 0.5) |
| | Policy update frequency | 2 |
| | Weight of BC term ($\alpha$) | 2.5 |
| | Weight of ISMAQ($\xi$) | 7 |
| | Number of ensemble-Q ($K$) | 10 |
| | Window for SMA ($w$) | 500 |
| | Reference SMA distance ($d$) | 500 |
| | Critic hidden dimension | 256 |
| | Critic hidden layers | 2 |
| | Critic activation function | ReLU |
| Network architecture | Actor hidden dimension | 256 |
| | Actor hidden layers | 2 |
| | Actor activation function | ReLU |

