# OpenReview forum: "Automatic Fine-Tuned Offline-to-Online Reinforcement Learning via Increased Simple Moving Average Q-value"
_ICLR.cc/2024/Conference — Submitted to ICLR 2024_

### Official Review · Reviewer_g2Zo · 2023-10-25

**Soundness:** 3 good
**Presentation:** 3 good
**Contribution:** 3 good
**Rating:** 6
**Confidence:** 4

**Summary:**

The paper deals with offline to online RL. A simple algorithmic novelty is presented and experiments are conducted to show that compared to alternative methods, especially at the beginning of online training, severe performance drops can be reduced.

**Strengths:**

* The paper is very carefully prepared and well written.
* The proposed algorithm is simple perhaps even elegant
* The results are promising

**Weaknesses:**

* The performance of the algorithm has not been studied on benchmarks with stochastic MDPs.

**Questions:**

No questions, but further comments:
* At “Batch or offline reinforcement learning methods”, I would think it would be good to also mention one of the older papers on batch/offline RL, e.g. [1], so that it is clear that the topic did not just come up in 2020. But I consider this a matter of taste.

* The term "policy collapse" remains too vague. It should be clarified what is meant by it.

* At “Many previous methods Zheng et al. (2023b); Lee et al. (2022); Zhang et al. (2023) could achieve better policies than their pre-trained offline models but suffer from policy collapse at the beginning of the transition from offline to online”, the reader might ask whether safe RL should solve the problem, and thus be mentioned, e.g., [2][3][4] and explained why these techniques cannot be used here.

* Furthermore, it seems useful to distinguish the current work from [5] and {6], which are also robust against performance losses in the online phase.

[1] Lange et al, Batch Reinforcement Learning, 2012\
[2] Laroche et al, Safe Policy Improvement with Baseline Bootstrapping, 2017\
[3] Nadjahi et al.: Safe policy improvement with soft baseline bootstrapping. 2019\
[4] Scholl et al.: Safe Policy Improvement Approaches and their Limitations, 2022\
[5] Swazinna et al., User-Interactive Offline Reinforcement Learning, 2022\
[6] Hong et al., Confidence-conditioned value functions for offline reinforcement learning, 2022


* In “We use a bootstrapped ensemble Q-network with an outlier filtering technique for more accurate value estimation to reduce the uncertainty encountered during the distribution drift”, it remains unclear which of this existed before and which is an innovation.

* The discussion of offline RL in Section 2.1 does not address model-based offline RL. I think it should be clarified that only model-free offline RL is considered in the paper.

* In Table 1, there is no mention of what the numbers behind the $\pm$ are. If nothing is mentioned, then they should be estimates of statistical uncertainty, e.g., the standard error. The standard deviation should never be used after a $\pm$ because it cannot serve as a measure of the uncertainty of the mean preceding the $\pm$. Finally, the uncertainty of the mean becomes smaller as the number of experiments increases, while the standard deviation does not. The authors should ensure that the standard error is used at this point, or another measure of uncertainty, such as the 95% confidence interval (but in this case it should be mentioned in the caption).

* In Table 1 there are some format errors, like "90.7 $\pm$ 2.00", where the number of decimal places of the uncertainty (2.00) does not match the number of decimal places of the measured value (90.7). So it must be 90.7 $\pm$ 2.0 or 91 $\pm$ 2.

* In the text "fine-tune" is used mostly, but in one place "finetune" is used.

* The following references are duplicated:

Philip J Ball, Laura Smith, Ilya Kostrikov, and Sergey Levine. Efficient online reinforcement learning
with offline data. arXiv preprint arXiv:2302.02948, 2023a.\
Philip J Ball, Laura Smith, Ilya Kostrikov, and Sergey Levine. Efficient online reinforcement learning
with offline data. arXiv preprint arXiv:2302.02948, 2023b.

Han Zheng, Xufang Luo, Pengfei Wei, Xuan Song, Dongsheng Li, and Jing Jiang. Adaptive policy
learning for offline-to-online reinforcement learning. arXiv preprint arXiv:2303.07693, 2023a.\
Han Zheng, Xufang Luo, Pengfei Wei, Xuan Song, Dongsheng Li, and Jing Jiang. Adaptive policy
learning for offline-to-online reinforcement learning. arXiv preprint arXiv:2303.07693, 2023b.

---

> ### Author Response · Authors · 2023-11-21
> **Response to Reviewer g2Zo**
>
> We appreciate your guidance and suggestions. Here are the responses:
>
> 1. We have selected Balanced Replay (Lee et. al. 2022) as one of the benchmarks using stochastic policy gradient. It is based on CQL
>     (Kumar et. al. 2020) derived from SAC (Haarnoja et. al. 2018).
> 2. We have added the citation of (Lange et. al. 2012) in the suggested sentence in the “Introduction” section.
> 3. We have added a detailed description of what “policy collapse” indicates in the “Introduction” section. It means the models could not
>     maintain the performance of existing pre-trained offline models. Instead, they suffer a sudden decrease in performance.
> 4. We have added the description of why Safe RL cannot be used here since the key limitation is that all these papers assume the ability
>      to interact with the environment and behavior policy to accurately estimate the baseline performance. And cited the papers
>      mentioned following the explanation of “policy collapse” in the “Introduction” section.
> 5. We have included [5] in the related work of offline-RL since [5] claims it is an offline method. And [6] is offline-to-online RL although it
>      mainly focuses on offline-RL it has online fine-tuning comparison with other algorithms. [6] mainly focuses on comparison with offline
>      methods. However, with online fine-tuning experiments, they focus on discrete tasks (Atari games).
> 6. We have clarified that the outlier filtering method is an innovation to our best knowledge. Since bootstrapping ensemble-Q is a
>      previously developed methodology.
> 7.  We have added a description for clarification that we mainly consider model-free methods in this paper (at the end of “Related Work”
>       – “Offline RL”).
> 8.  We have removed all the ± representations. Following previous studies: CQL, D4RL, and so forth.
> 9.  We have fixed the precision consistency in Table 1.
> 10. We have fixed the inconsistency using “finetune” instead of “fine-tune”.
> 11. We have removed the duplicated citations.

---

> > ### Comment · Reviewer_g2Zo · 2023-11-21
> > **Question**
> >
> > I appreciate the authors' response and will try to provide feedback before the discussion phase ends. First of all, a question: to which comment does the sentence `We have selected Balanced Replay (Lee et. al. 2022) as one of the benchmarks using stochastic policy gradient. It is based on CQL (Kumar et. al. 2020) derived from SAC (Haarnoja et. al. 2018).`   refer?

---

> > > ### Author Response · Authors · 2023-11-22
> > > **Response to Question about Stochastic policy gradient**
> > >
> > > This comment refers to the weakness commented ("The performance of the algorithm has not been studied on benchmarks with stochastic MDPs.")
> > >
> > > Based on the description of MuJoCo environments. https://gymnasium.farama.org/environments/mujoco/
> > > They are stochastic, thus, we assume the reviewer was referring to stochastic policy gradient methods.
> > >
> > > If misunderstood please let us know. Thank you and sorry about the confusion.

---

> > > > ### Comment · Reviewer_g2Zo · 2023-11-22
> > > > **Stochastic MDPs**
> > > >
> > > > Thanks for the clarification. That was indeed a misunderstanding, because I really meant stochastic MDPs, i.e. MDPs with stochastic state transitions. The MuJoCo benchmarks only use deterministic state transitions. The stochastic selection of initial states is not sufficient for this. In my opinion, it is currently a problem that stochastic MDPs are not sufficiently represented in D4RL, so it is to be feared that some algorithms will not work well in stochastic MDPs, which can also contain multimodal probability distributions. I would like to refer to (S. Mannor and A. Tamar, Towards Deployable RL -- What's Broken with RL Research and a Potential Fix, 2023).
> > > > Regarding the present paper, I can say that I think it is good that it is already pointed out in the abstract that D4RL is used. Additional experiments with stochastic MDPs would, in my opinion, significantly enhance the paper. As it is, I think the limitation that the new algorithm has not been validated for stochastic MDPs should be pointed out somewhere in the text.

---

> > > > > ### Author Response · Authors · 2023-11-23
> > > > > **Response to "Stochastic MDPs"**
> > > > >
> > > > > Thanks for the clarification. The lack of evaluation on stochastic MDPs is described as a limitation in the discussion section.

---

> > ### Comment · Reviewer_g2Zo · 2023-11-22
> > **Mal-formated reference**
> >
> > The reference to "Safe Policy Improvement with Baseline Bootstrapping
> > Romain Laroche, Paul Trichelair, Remi Tachet Des Combes Proceedings of the 36th International Conference on Machine Learning, PMLR 97:3652-3661, 2019" is broken and needs some tuning.

---

> ### Author Response · Authors · 2023-11-22
> **Response to "Mal-formated reference"**
>
> Thanks for the correction. We have fixed the citation and have uploaded the revised version.

---

### Official Review · Reviewer_AHJ2 · 2023-10-26

**Soundness:** 1 poor
**Presentation:** 2 fair
**Contribution:** 1 poor
**Rating:** 3
**Confidence:** 5

**Summary:**

This manuscript proposes a new method – Increased Simple Moving Average of Q-value (ISMAQ) for the fine-tuning problem in offline-to-online RL. Although the proposed formulation seems interesting and novel, the reviewer believes this manuscript still has room for improvement before publishing. See details below.

**Strengths:**

1. The observation and insight in Figure 1 seem interesting – where the Q-mean of non-expert data increases while the Q-mean of expert decreases.
2. The proposed method in equation 2 seems to be novel.

**Weaknesses:**

1. [Minor] The citation format needs to be updated. Please use `\citep` instead of `\cite`, when the papers being cited are not used as nouns. For example, in the second line of the introduction, Silver et al. (2017), and the other citations should appear as (Silver et al., 2017). There are many other citations that use the wrong format, which inevitably affects the readability of the manuscript.
2. [Minor] The citation in Section 2.2.1 seems weird – it contains two papers by Ball et al (2023a;b), which appear to be the same paper in the reference.
3. [Major] While the observation in Figure 1 of Section 4.1.1 seems interesting, the reviewer does not fine the conclusions convincing enough, since it only has experimented with `halfcheetah`. The reviewer is expecting more environments from D4RL (such as `walker`, `hopper`, `ant-maze`) and even other environments such as `adroit-hand`, as adopted by the Cal-QL [1] paper.
4. [Major] The Lemma in Section 4.1.2 is not informative enough – the reviewer does not understand what is the Lemma proving. The reviewer is expecting a rigorous lemma to be written with math notations, not text descriptions. Based on the current presentation of the Lemma, the reviewer cannot tell the correctness of the lemma.
5. [Major] The experiments in 3 are also not conclusive enough using only the `walker2d` environments. The reviewer understands the authors’ motivation is to provide a justification for the ReLU operator, but the reviewer is expecting more experiments in other environments as suggested in 3.
6. [Minor] The original paper of REDQ [2] is by Chen et al., not Zhao et al. The reviewer understands that Zhao et al., proposed a new method for offline-to-online that is built on top of [2], but it might be better for the authors to clarify the origins of REDQ [2] in Section 5.1.
7. [Major] For the experimental evaluations in Section 5, only conducting experiments in D4RL locomotion is not enough. The reviewer is expecting more environments (see e.g., [1]).

**Questions:**

[Major] At the bottom of Section 4.2, the authors claim that

> These two techniques both could be implemented with minimal changes with only $\mathcal{O}(1)$ time complexity…

What does “These two techniques” refer to? Since the author also mentioned $\mathcal{O}(1)$ complexity in the abstract, the reviewer would expect more discussion on where the $\mathcal{O}(1)$ complexity comes from and how it is achieved.

---

> ### Author Response · Authors · 2023-11-21
> **Response to Reviewer AHJ2**
>
> We appreciate your guidance and suggestions. Here are the responses:
>
> 1. We have modified the citation format and uploaded the revised paper.
> 2. We have removed the redundant citation and uploaded the revised paper.
> 3. We have improved the Lemma and its proof with more
> 4. We have changed the naming from REDQ to REDQ+ABC to avoid confusion.
> 5. We have added the partial experiment results of AntMaze and Adroit tasks, please refer to Appendix D.3. (Experiments of Balanced
>     Replay are still running, we will provide an updated result once it has been completed.)
> 6. The two techniques refer to CER (Combined Experience Replay) and deleting the oldest policy, we have modified the wording to avoid
>     confusion.

---

### Official Review · Reviewer_Thik · 2023-10-31

**Soundness:** 3 good
**Presentation:** 3 good
**Contribution:** 3 good
**Rating:** 6
**Confidence:** 4

**Summary:**

The paper proposes a novel regularization strategy that should help to efficiently adapt offline pre-trained policies with additional online data in an offline-to-online setting. Specifically, the authors propose an exploration bonus to add on top of previously existing offline RL algorithm TD3+BC in order to not remain too conservative when moving from offline to online training. The new term is based on the difference between the average Q-value of the current and a reference time step. Additionally, low-complexity buffer techniques are incorporated into the method in order to adapt to distribution drift due to the changing policy.

**Strengths:**

Many offline RL algorithms as well as exploration techniques focus in one way or the other on a measure of uncertainty for their regularization - offline RL approaches penalize it to remain within data support, while exploration schemes explicitly seek out uncertainty for information gain. The proposed technique however uses nothing of the sort, instead it simply measures the difference in average episodic Q-values over training time. Based on the observation that these values increase only for sub-optimal agents & decrease over time when the buffer only contains expert data, it is used as an additional loss term to improve the policy in the online training part of the algorithm. The method is very simple and does not require training of additional models like most other regularization schemes in this context. At the same time it appears to work well and to the best of my knowledge it can be considered novel. Also, the outlier filtering appears to be an innovative concept to improve the stability of the method.
Furthermore, the empirical performance on last-10 appears to match the prior SotA, while it outperforms the prior best on first-10.

**Weaknesses:**

It is a little unclear to me what exactly first-10 & last-10 performance means (I may have missed it) - if it refers to the average return of the policy during the first and last 10 gradient updates, I am wondering whether the comparison for the first-10 case is meaningful: From the appendix I gather that f=2, i.e. the policy is updated every 2 steps, so you only have really 5 different policies. Also, the algorithm starts with the offline pre-trained policies, which we know perform well since TD3+BC is known to work on the presented datasets. Is it possible that 5 policy updates is just too little to move far away from this & that is the reason why it is that good? I know that prior O2O approaches had trouble to even maintain the offline performance when they moved to the online phase, however it seems odd that others (like e.g. TD3+BC to TD3) basically drop immediately by a huge margin. Do they all start with the same pre-trained policy performance? What do you attribute this difference especially during the first few updates too? I would suggest to extend the plots towards the left so that one can also see the offline training phase and directly inspect what happens when you move from offline to online. The last-10 performance isn't really better than the prior SotA by REDQ, so since the main contributions are novelty and first-10 performance, I think it is important to examine the latter more closely.

I believe some other prior works should also be considered in the related work section:

[1] Ghosh, D., Ajay, A., Agrawal, P., & Levine, S. (2022). Offline rl policies should be trained to be adaptive. ICML 2022

[2] Hong, J., Kumar, A., & Levine, S. (2022). Confidence-Conditioned Value Functions for Offline Reinforcement Learning. ICLR 2023

[3] Swazinna, P., Udluft, S., & Runkler, T. (2022). User-Interactive Offline Reinforcement Learning. ICLR 2023

They are also concerned with offline to online learning, just that their online phase is a little shorter and their adaptations thus look a little different than the one you consider. Still, when thinking about O2O they are closely related and should be considered.

**Questions:**

I do not understand figures 7/8:
- what is the middle figure showing - there is no legend so it's unclear which of the other two legends is active here?
- since the colours are the same in each graph, it is a bit misleading what this means
--> e.g. is the blue one a combination of the legends (ISMAQ weight=1 AND K=5)?

what does no_ismaq mean in fig.10? The text says something about plainly using Eq. 5, but there the ISMAQ weight is already contained...

in fig 9 you evaluate different choices for d - have you tried really small ones as well, like 1? I mean at some point it has to collapse right?

---

> ### Author Response · Authors · 2023-11-21
> **Response to Reviewer Thik**
>
> We appreciate your guidance and suggestions. Here are the responses:
>
> 1. The first-10 indicate the first 10 evaluations (where each evaluation is averaged with 10 episodes) and the evaluation frequency is
>     1,000 timesteps. We have listed the details of the experiments in Appendix A - “Training and Evaluation details”.
> 2. We keep the same settings as TD3 and TD3+BC, where the policy update frequency is 2. It implies that for every 2 timesteps of critic
>     training, the actor will only be updated once. We have K ensemble-Q models (K=10) and only one actor (one policy) in our ISMAQ.
> 3. The significance of comparing TD3+BC to TD3 is that we use the same set of pre-trained offline models, neither of these two changes
>     the offline training algorithms. The abrupt drop in performance for TD3+BC to TD3 results from the fact that it removes the behavior
>     cloning term in Eq.1 once it enters the online fine-tuning mode.
> 4. Our method uses the pre-trained TD3+BC models as the starting point. Thus it will be fair to compare the initial performances between
>     ours and TD3+BC to TD3. For other methods (except Balanced Replay uses CQL) they re-train the offline models. Thus, they are
>     expected to have better initial pre-trained offline results since they have improved offline training. We have added Table 2 to
>     demonstrate the pre-trained offline model performance in Appendix A.1.
> 5. Maintaining the performance of pre-trained offline models is critical for offline-to-online RL models. Otherwise, we could just train an
>     online model from scratch. That is the main reason we include the first-10 evaluations as a metric to compare the models.
> 6. We have added the mentioned articles in the “Related Work” - “Offline RL” section since all of them focus on pure offline settings
>     [1] proposes that offline RL methods should instead be adaptive in the presence of uncertainty to solve POMDP
>     [2] mainly focuses on comparison with offline methods. However, online fine-tuning experiments, focus on discrete tasks
>          (Atari games).
>     [3] focuses on user adaption, the methods we have compared focus on auto-tuning or hyperparameter tuning.
> 7. We have fixed the formats of Figures 7 & 8 and have uploaded the revised paper.
> 8. We have revised the description of “no_ismaq”, it contains all other techniques (ensemble and CER) but not the added policy
>     regularization term.
> 9. Since the distance to the reference point might vary, the reference value is also calculated with SMA. Thus, the current value and the
>     reference value are both Q-SMA. Both of them are stable with an optimized window size for SMA. Hence, the adjustment of the
>     hyperparameter d will not affect the model performance significantly. We have added this description to the paper.

---

### Official Review · Reviewer_74i3 · 2023-11-01

**Soundness:** 2 fair
**Presentation:** 1 poor
**Contribution:** 2 fair
**Rating:** 3
**Confidence:** 4

**Summary:**

The paper proposes an offline-to-online RL algorithm named ISMAQ (Increased Simple Moving Average of Q-value). This method extends TD3+BC and introduces a new loss term into the actor loss, designed to selectively raise the average Q-values based on convergence. Additionally, it incorporates various techniques, including critic ensemble, outlier filtering, combined experience replay, and the removal of the oldest transition in the buffer. In the experiment, ISMAQ outperformed several previous methods on the D4RL locomotion benchmark.

**Strengths:**

1. The proposed method builds upon TD3+BC and improves over it on both offline-to-online setting and online from scratch setting.

2. The ablation studies testing the sensitivity of each component in Sections 5.3 and 5.4 are informative.

3. The paper studies an interesting and important problem.

**Weaknesses:**

1. The comparisons with several other offline-to-online RL algorithms are missing [1, 2, 3, 4]. Several of them are missing in the related work as well.

2. The method is only evaluated on D4RL locomotion tasks. It would be beneficial to include results on the D4RL Antmaze tasks as in [2,3,4] and the Adroit binary task as in [1, 4] which require higher sample efficiency than the locomotion tasks.

3. I don’t think the following sentence is true. Does AWAC need either of the requirements?
>Unfortunately, the aforementioned offline-to-online methods need at least one of the following requirements that makes them resource-consuming Yu & Zhang; Zhao et al. (2022); Lee et al. (2022); Nair et al. (2020); Luo et al. (2023): Changing the offline training processes (requires re-training of the offline models), introducing additional models other than existing ones, and maintaining multiple buffers.

4. The REDQ (Zhao et al. 2022) baseline is confusing to me. The original REDQ should be the paper [5]. I would suggest changing the name of that baseline.

5. Is the following statement in  Section 4.3 correct? I don’t think any of [1, 2, 3, 4] is using ensemble.
> almost all previous O2O studies take advantage of certain kinds of ensemble learning

5. Many citations are styled incorrectly and are difficult to read – \citep{} should be used instead of \citet{}. See the official formatting instructions below. Additionally, several papers are cited multiple times, such as Ball et al. (2023a and 2023b) and Zheng et al. (2023a and 2023b).
>When the authors or the publication are included in the sentence, the citation should not be in parenthesis using \citet{} (as in “See Hinton et al. (2006) for more information.”). Otherwise, the citation should be in parenthesis using \citep{} (as in “Deep learning shows promise to make progress towards AI (Bengio & LeCun, 2007).”)

6. The style of Figures 7 and 8 is broken. They override the text above, and it's also confusing that there are three figures accompanied by only two captions.



[1] Nair et al., AWAC: Accelerating Online Reinforcement Learning with Offline Datasets, 2020

[2] Zheng et al., Online Decision Transformer, 2022

[3] Wu et al., Supported Policy Optimization for Offline Reinforcement Learning, 2022

[4] Nakamoto et al., Cal-QL: Calibrated Offline RL Pre-Training for Efficient Online Fine-Tuning, 2023

[5] Chen et al., Randomized Ensembled Double Q-Learning: Learning Fast Without a Model, 2021

**Questions:**

1. How is the offline pre-training phase performed? Is the proposed method simply pre-trained using TD3+BC, or is the ISMAQ loss also combined?

2. In Figure 12, it appears that the offline pre-trained performance of ISMAQ and TD3+BC should be comparable. However, in Figure 5, there is a significant difference in the initial performance between ISMAQ and TD3+BC_to_TD3. What’s the reason for this discrepancy?

---

> ### Author Response · Authors · 2023-11-21
> **Response to Reviewer 74i3**
>
> 1. For the comparison with related works:
>      a. AWAC [1] is using offline data to accelerate online learning. Its goal differs from our assumption where we start from an existing
>          pre-trained offline model. Since in many real-world problems we do not have simulators to train online models from scratch and/or
>          training from random policies is infeasible/risky. For example, disease treatment, trading, and so forth.
>      b. We did not include Online Decision Transformer [2] because none of their medium and medium-replay tasks could reach 100.0 (the
>           normalized expert scores), on the other hand, the methods in our paper (TD3+BC to TD3) outperform in these tasks.
>      c. [3] considers itself as an offline-RL method, and was compared with offline-RL in their paper.
>      d. Cal-QL[4] is categorized as an offline-to-online RL method. We have tried to run the experiments with it. However, the CUDA and
>          CuDNN versions on our server are incompatible with their requirement since they are using Jax. We will try to update the result if
>          the problem is resolved.
> 2. We have added the partial experiment results of AntMaze and Adroit tasks, please refer to Appendix D.3. (Experiments of Balanced
>      Replay are still running, we will provide an updated result once it has been completed.)
> 3. We have removed the citation of AWAC in the offline-to-online requirement section.
> 4. We have modified the naming of “REDQ” to “REDQ+ABC” in this paper to avoid confusion.
> 5. We have modified the wording in Section 4.3 from “almost all previous O2O studies take advantage of certain kinds of ensemble
>     learning” to “many previous O2O studies take advantage of certain kinds of ensemble learning”
> 6. We have fixed the citation styles.
> 7. We have fixed Figures 7 & 8 to avoid confusion.
> 8. Our method ISMAQ uses the pre-trained TD3+BC model and then does the online fine-tuning with the added policy regularization
>    term. We have added a more detailed description of the methodology in Section 4.1.3 - “ISMAQ Implementation”
> 9. In Figure 12, we incorporate the ISMAQ term with TD3+BC in pure offline training mode. However, in Figure 5, ISMAQ and TD3+BC to
>     TD3 are both initialized with pre-trained TD3+BC. Where TD3+BC to TD3 naively converts from offline mode to online mode with the
>     weight of behavioral cloning term changed from ⍺=0.4 to ⍺=0. We will add this explanation to the paper.

---

> > ### Comment · Reviewer_74i3 · 2023-11-23
> >
> > I thank the authors for providing the explanations to my questions.
> > ___
> >
> > > **AWAC [1] is using offline data to accelerate online learning. Its goal differs from our assumption where we start from an existing pre-trained offline model.**
> >
> > I believe the authors have some misunderstanding about AWAC. AWAC is an offline-to-online RL method, as shown in Figure 2 of its paper. Therefore, I still recommend that the authors include it as a baseline method.
> > ___
> >
> > > **We have added the partial experiment results of AntMaze and Adroit tasks, please refer to Appendix D.3.**
> >
> > The proposed method, ISMAQ, appears to perform poorly in the newly added tasks, such as achieving a score of 0.0 in antmaze-large-diverse.
> > ___
> >
> >
> > Overall, as the paper still lacks some important baselines and does not seem convincing enough in the newly added benchmark tasks, I will maintain my current score.

---

> > > ### Author Response · Authors · 2023-11-23
> > > **Response to newly adde comment**
> > >
> > > Thanks for the comment. We will keep improving our algorithm and evaluation.

---

### Meta-Review · Area_Chair_gaL6 · 2023-12-09

**Metareview:**

This paper proposes a method to augment the policy update of actor-critic algorithms when starting to fine-tune from pre-trained checkpoints obtained by offline RL algortihms to allow for better performance. Unfortunately, I have several concerns with this paper, which include many of the points that the reviewers make:

- First, the theoretical result is not rigorous, without any symbols, which nullifies its importance totally.

- While I understand that the goal of the paper is to develop an approach to make it possible to fine-tune pre-trained checkpoints (and this is a bit different from conventional offline-to-online fine-tuning which retains offline data for training or modifies offline training algorithms for making it possible to do online fine-tuning), it is important to compare basic baselines: running TD3+BC for longer, instead of TD3 during online learning; O3F which adds exploration to online learning (Mark et al.); methods in the reincarnating RL paper (Agarwal et al. 2022).

Additionally it is also important to discuss the comparison to offline-to-online settings that people typically study and evaluate methods from the online fine-tuning literature as reference (e.g., Cal-QL, AWAC, IQL)

- The writing needs a lot of improvement, and the method needs to be presented coherently in one place.

I would encourage authors to address the above points and the reviewers concerns (most of which I agree with), and submit to the next conference.

**Justification For Why Not Higher Score:**

Several concerns as listed above

**Justification For Why Not Lower Score:**

N/A

---

### Decision · Program_Chairs · 2024-01-16

Reject